# The Influence of Media Exposure on Anxiety and Working Memory during Lockdown Period in Italy

**DOI:** 10.3390/ijerph18179279

**Published:** 2021-09-02

**Authors:** Rosa Angela Fabio, Rossella Suriano

**Affiliations:** Department of Clinical and Experimental Medicine, University of Messina, 98100 Messina, Italy; suriano.rossellaz@gmail.com

**Keywords:** COVID-19, social distancing, subjective loneliness, media exposure, anxiety, working memory

## Abstract

The rapid spread of the coronavirus pandemic has caused anxiety around the world. During lockdown, the media became a point of reference for people seeking information. However, little is known on the relationships between anxiety resulting from persistent media exposure to coronavirus-related programs and the effects produced on working memory. In this work, a total of 101 Italian citizens (53.7% female) aged between 18 and 45 years old, who were from 14 provinces in Italy, participated in an online survey. Participants were presented with media exposure and anxiety questionnaires and they were instructed to carry out working memory tasks (visual and auditory n-back). The results showed that media exposure is related to anxiety. It was also found that high levels of anxiety have a negative influence on the performance of both visual and auditory working memory tasks in terms of increased reaction times of responses and decreased accuracy. The results were critically discussed in the light of the Social Compensation Hypothesis.

## 1. Introduction

From 11 March 2020, to counter the spread of the new coronavirus, a decree was issued in Italy (DPCM 11 March 2020) which, among other measures, foresaw the obligation for all people to stay at home unless there were valid reasons not to do so. For the first time in the history of the Italian Republic, such an important measure was taken, which influenced citizens’ social relationships. Especially at the psychological level, the consequences of both real and perceived isolation were documented [1,2,3,4]. Cross-sectional studies were conducted in Europe to examine the relationships between the fear of contagious diseases and other factors. The results showed that neuroticism, age and sense of belonging to the country did predict fear of contagious disease [5]. Moreover, in this context, mass media had a central role: since the start of the lockdown and the continuing of the DPCMs, there has been an 87% increase in the use of mass media amongst the general population, and in a specific way, the use of chats has considerably increased [6]. A study conducted during the lockdown on a sample of adults showed that, on average, smartphones were used for a period of about 45 min longer than in the period before the pandemic. The most interesting aspect was that this increase was attributable to longer durations of calls and messaging [7].

### 1.1. Social Distancing, Subjective Loneliness and Media Exposure

The Social Compensation Hypothesis (SCH) [8] offers a way to conceptualize the increase in Internet use, particularly during times of stress or crisis. The coronavirus pandemic can be considered a crisis, understood as a set of specific and surprising events, which are perceived as a serious threat and produce high levels of uncertainty [9]. According to this theory, the increase in interactions through the media compensates for the reduction in interactions in person. Moreover, as pointed out by the theory of media dependency [10], it is precisely during serious social disturbances that a strong need emerges to find information, as well as to maintain and strengthen interpersonal relationships in the name of comparison and mutual support [11]. Therefore, it is during a time of crisis that people tend to increase their dependency on media, manifested in the insistent and continuous search for increasingly accurate and updated information, in order to make adequate decisions on protective behaviour. While numerous studies showed that the high frequency and long duration of face-to-face social interactions resulted in lower levels of loneliness [12,13,14], there is conflicting evidence of the extent to which communication via digital tools, and therefore online interactions (video calls, messages, e-mails), can reduce loneliness too. A recent study [15] examined whether individuals who were experiencing high levels of loneliness during the forced isolation due to the COVID-19 pandemic were more prone to feeling anxious, and whether their sense of loneliness prompted excessive social media use. Their findings suggest that isolation probably reinforced the individuals’ sense of loneliness, strengthening the need to be part of virtual communities. On the other hand, the facilitated and prolonged access to social media during the pandemic risked further increases anxiety, generating a vicious cycle that, in some cases, may require clinical attention [15]. In this context, the coronavirus pandemic represents an unusual opportunity to investigate whether communication via the media can effectively compensate for face-to-face interactions [16]. Although the psychological impact of lockdown and quarantine on people was previously documented [17], the implications of using the media during the coronavirus pandemic on psychological functioning are still unclear. Some recent research suggests that the media can cause a high level of stress in most people [18,19]. However, only two studies analysed the correlation between mass media exposure and COVID-19 on mental health. The results of these studies suggest that frequent and repeated exposure to mass media is significantly positively correlated with symptoms of stress and anxiety within the general population in China and Germany, respectively [20,21].

### 1.2. Anxiety and Working Memory

Moreover, it was shown that anxiety influences cognitive performances [22], and mainly the working memory (WM), a temporary storage system with limited capacity that allows the active representation and manipulation of information within a short period of time [23,24]. It is important to clarify that not all of the studies that investigated the relationship between anxiety and WM reported consistent results. Some researchers proposed that anxiety may decrease attentional control and executive processes, and thus impair the ability to maintain relevant information and inhibit irrelevant information [25,26]. Other researchers confirmed these results as both anxiety and WM rely on prefrontal and parietal regions in the brain, competing for limited neural resources [27]. Results from other researchers did not show a significant correlation between anxiety and WM [28,29], but instead, some studies found that anxiety improves WM performance [30,31]. Another component that has a significant impact on WM performance, directly and/or indirectly, is mass media exposure [32,33]. However, very little is known regarding the actual effects on cognitive processes brought about by the high levels of anxiety caused by high mass media exposure.

### 1.3. The Current Study

In the current study, the relationships between media exposure, state anxiety, and visual and auditory WM during the COVID-19 pandemic, in a sample of healthy Italian adults, were examined. More specifically, the purpose of this study was twofold: (a) to evaluate the influence of high levels of media exposure and COVID-19 media exposure on levels of anxiety and (b) to evaluate the effects of high levels of anxiety on the performance of both visual and auditory WM tasks. The present study hypothesized that high levels of media exposure may increase levels of state anxiety, and that high levels of state anxiety may impair WM performance.

## 2. Materials and Methods

### 2.1. Participants

The participants of this study were recruited between 1 May 2020 and 30 June 2020 through social networks such as Facebook and/or Instagram, where a questionnaire was shared. After completing it via the Google Forms platform, they were contacted and invited to collaborate in the study by carrying out the proposed activities. In total, the initial sample included 173 subjects, but only 101 participants agreed to join in the shared video call and to perform the WM tasks. The final sample included 47 men (46.5%) and 54 women (53.5%). Their ages ranged between 18 and 45 years, with an average age of 31.81 years (SD = 13.02). Subjects came from all over Italy; specifically, 63.37%, or the majority, came from Southern Italy, 30.7% from Northern Italy and 5.94% from the Central Italy. As regards level of education, 60.4% of the participants had obtained a high-school diploma, 30.7% at least a degree, 7.92% a middle school certificate and only 0.99% had an elementary school certificate. Out of 101 people, 67 said they had not worked during the coronavirus pandemic, while 34 continued their work, some of them working from home and only a few in person. In fact, 20.79% of the sample declared they had never left their home during the lockdown, and 61.38% went out only a few times out of necessity, while only 17.82% said they had left their home more than 5 times a week (for example, policemen and physicians). The sample is representative of a wide range of professional occupations, divided into three distinct levels based on the activity that most exposed them to social contact. More precisely, the first level included the 14.85% of the sample that corresponded to the categories most exposed to infection, as they were more physically involved: healthcare personnel, law enforcement and food sales staff. The 33 public employees, teachers and managers of some activities (11.88%) were included in the intermediate level, who alternated their presence according to work needs. The third level included 73.27% of the participants (38.6% of whom were students) who, being able to carry out their activities at their homes, were less socially exposed. These included freelancers, housewives, some traders and the unemployed. The results showed that 9.9% of the sample spent lockdown alone, 22.8% lived with only one person and 67.32% with more than 2. With an α ≤ 0.05, and a power ≥ 0.80, a sample of n = 98 participants was needed to achieve a power for differences. Given these analyses, the final sample of N = 101 was adequate.

### 2.2. Procedure

In order to start the procedure and, therefore, recruit participants, a post was published on various social networks (such as Facebook and Instagram) that aimed to clearly explain the methods of participation in the research, and also asked for an availability of between 35 and 40 min. Each individual participant voluntarily agreed to participate in this research study. Only after providing informed consent were users initially directed, by a link, to the Google Forms platform for the completion of a series of questionnaires aimed at evaluating various dimensions, including social distancing and isolation, media exposure and state anxiety. Only in a second session (the day after), and by agreement, were the participants contacted and invited to a video call via platforms such as Skype, Teams or Zoom for the evaluation of WM performance. After sharing the screen, they were provided with the necessary information to connect to the internet page on the website https://new.cognitivefun.net/ (accessed on 25 April 2020) where they would find the visual n-back and auditory n-back tests.

### 2.3. Measurement

Both self-report tools and performance tests were used in the present study. In particular, two questionnaires, the Social distancing and Subjective loneliness (SDSLQ) and the Media exposure questionnaires (MEQ), were adapted for the evaluation of real isolation, subjective isolation and exposure to the media [34,35]. Anxiety levels were measured through the administration of the standardized STAI test [36], while the performance of visual and auditory WM was investigated through the n-back paradigm (https://new.cognitivefun.net/ accessed on 25 April 2020).

#### 2.3.1. Social Distancing and Subjective Loneliness Questionnaire

In order to obtain a research tool that would allow the estimation of both real social distancing and the sense of subjective loneliness, a questionnaire was created that comprised two scales, each of which was made up of 3 items [34]. The first investigated social isolation, which is a condition that precludes the lack of people on a physical level. For example, a proposed item was “How often have you met people in the last week?”. The second scale referred to a feeling of loneliness. An example of an item was “How often do you feel the lack of company?”. Participants filled out each item with the 4 points of the Likert scale, from 1 (never) to 4 (always). Verified internal agreement of both scales was carried out through Cronbach’s alpha coefficient: the former, regarding social distancing, was α = 0.82, while, for the latter scale that was related to subjective loneliness, it was equal to α = 0.78.

#### 2.3.2. Media Exposure Questionnaire

To assess exposure to media content related to COVID-19, a questionnaire was adapted comprising 13 items aimed at investigating the intensity of use of the media during the COVID-19 lockdown [35]. Additionally, in this case, subjects were asked to provide feedback on the intensity and the response mode was a 4-point Likert scale, from 1 (never) to 4 (always). From the analysis of the internal consistency, Cronbach’s alpha was α = 0.81.

#### 2.3.3. Anxiety Questionnaire

The STAI-Y (State-Trait Anxiety Inventory) is an easy-to-apply-and-interpret tool that is used to detect and measure anxiety [36]. It includes two scales (Y1 and Y2), each made up of 20 items that, respectively, evaluate state anxiety, where the emotional condition of the subject at the time of administering the questionnaire is identified, and trait anxiety, which investigates how the subject feels habitually [37,38]. It is thus possible to make a distinction between anxiety understood as a symptom and anxiety expressed as a habitual way of responding to external stimuli [38]. The first refers to a situational activation, a temporary condition referring to a well-defined moment. The second refers to a persistent emotional state and can be considered as a relatively stable characteristic of the personality. Each item of the questionnaire presents a statement to which the subject must respond in terms of intensity on a 4-point Likert scale: from 1 (not at all) to 4 (very much). Participants were asked to mark the number corresponding to their emotional condition. Higher scores correlated with higher levels of anxiety. The two scales, Y1 and Y2, can be used independently of each other, providing two separate and distinct results. The aim of this study was to evaluate the anxiety level of the participants in relation to a specific period, during the coronavirus pandemic. For this purpose, it was considered appropriate to administer only the STAI Y1.

#### 2.3.4. Visual and Auditory Working Memory Measurement

In order to evaluate the performance of WM, n-back was used [39], a continuous performance activity commonly used in the field of psychology and cognitive neuroscience. This paradigm has been widely used in the literature and has shown good psychometric properties [40,41]. The n-back task involves the serial presentation of stimuli (for example, an image or a sound), separated from each other by a few seconds. The participant has to decide whether the current stimulus matched that shown in previous steps. The n indicates the load factor, a variable number that can be adjusted up or down, respectively, to increase or decrease the cognitive load and to make the task more or less difficult [42]. This activity involves the active part of WM, as it requires the maintenance and continuous updating of information. In this study, the 2-back versions of both visual and auditory activity were used. It was explained to participants that, after starting a succession of 50 images from the start, they had to click on the figure that appeared two positions before. For example, if the sequence was machine–heart–machine, it was necessary to click on this last image since it had appeared two positions before. Each image appeared on the screen for 500 ms, followed by a screen that remained blank for another 3000 ms. The participant had 3500 ms, from stimulus onset until the beginning of the subsequent trial, to press the space bar. The https://new.cognitivefun.net/website (accessed on 25 April 2020) was available for 175 s for each of the two tests. The average completion time for participants was from 120 to 175 s for the visual n-back and from 112 to 175 for auditory n-back tests.

Later, when introducing the auditory n-back test, it was explained to the participants that it was necessary to proceed similarly to the previous activity, but instead of the 50 images, they would hear a sequence of 50 sounds; then, they had to click the space bar when they heard the specific sound of two positions before.

### 2.4. Statistical Analysis

To analyze the results, SPSS 24.0 statistical software was used (SPSS Inc., Chicago, IL, USA). Descriptive statistics (with means and standard deviations) were presented for each variable. The Pearson correlation analysis was conducted between the anxiety status, social distancing, subjective loneliness and media exposure questionnaires. Bonferroni’s correction was applied for multiple comparisons. In order to verify the causal relationship between media exposure as a predictive variable and anxiety as a dependent variable, and between social distancing and subjective loneliness and COVID-19 media exposure, linear regressions were conducted. In this paper, path analysis was applied to study the correlations and causations of anxiety status, social distancing, subjective loneliness and media on the parameters of accuracy and reaction time of WM.

### 2.5. Ethical Consideration

The study protocol was reviewed and approved by the institutional review board (IRB No. 2019-21-45) of the University of Messina. Potential study participants were provided with a detailed description of the study and were assured of confidentiality. Written, informed consent was obtained from each participant. They were also informed of the voluntary nature of the study participation and completion without any negative consequences.

## 3. Results

Table 1 shows the means and standard deviations of the anxiety status, social distancing, subjective loneliness and media exposure questionnaires. Table 2 shows the Pearson’s correlations between media exposure, COVID-19 media exposure, subjective loneliness, social distancing and anxiety status. As can be seen, media exposure is highly correlated with both COVID-19 media exposure and loneliness. COVID-19 media exposure is also related to anxiety and subjective loneliness. These results show that continuing to stay informed does not lead to anxiety reduction; on the contrary, it may amplify it. Moreover, high levels of social distancing do not show any correlation with anxiety and media exposures. High levels of subjective loneliness, instead, show correlations with both anxiety and media exposure, meaning that it is not living in objective social distancing situations that generates anxiety, but rather how those situations are processed (perceived loneliness).

### 3.1. Media Exposure and Anxiety

In order to verify the causal relationship between media exposure and COVID-19 media exposure on the levels of anxiety, a linear regression was conducted, considering media exposure as a predictive variable and anxiety as a dependent variable. The data were highly significant: the β index was equal to 0.35, t = 3.74, and *p* < 0.001. This result suggests that media exposure has a percentage of explained variance in predicting anxiety that is equal to 12% (coefficient of determination Rsquare = 12%).

Considering anxiety as an independent variable and perceived loneliness as a dependent variable, the β index was equal to 0.66, t = 8.92, *p* < 0.001, and the Rsquare value was equal to 50%; considering social distancing as the independent factor and anxiety as the dependent factor, β was not significant.

### 3.2. Anxiety and Working Memory

To evaluate the influence of high levels of anxiety on the performance of both visual and auditory WM, Table 3 shows the means and standard deviations for each parameter and Table 4 shows Pearson’s correlations of anxiety and WM measurements.

As can be seen from Table 4, anxiety is not significantly correlated with the correctness of responses, with visual n-back performance, or with auditory n-back performance. A correlation emerged between anxiety and reaction times in both n-back tests (respectively, r = 0.596, *p* < 0.001 and r = 0.244, *p* < 0.001). These results highlight the negative influence of high levels of anxiety on the performance of WM in terms of reducing response times, while they do not influence the correct responses.

Path analysis was performed to estimate the links between variables and to provide information on the underlying causal processes. The first was related to the reaction time of WM. From the analyses carried out, not all of the hypothesized links emerged as significant; therefore, some of them were eliminated, such as social distancing and media exposure.

The path diagram presented in Figure 1 shows that there is a direct and reciprocal relationship between anxiety and COVID-19 media exposure as well as between anxiety and subjective loneliness. With reference to the standardized total effects, the influence of anxiety on subjective loneliness is greater (β = 0.902, *p* < 0.001) than that on COVID-19 media exposure (β = 0.404, *p* < 0.007). Similarly, subjective loneliness also has a greater positive effect on anxiety (β = 0.77, *p* < 0.0001) than media exposure (β = 0.17, *p* < 0.02). It also emerged that anxiety has direct effects on reaction times both in visual and auditory n-back tasks, albeit to a different extent (respectively, β = 0.432, *p* < 0.002 and β = 0.33, *p* < 0.04).

The path diagram of Figure 2, related to correct WM responses, shows the direct and reciprocal relationship between the same variables (anxiety, COVID-19 media exposure and perceived isolation). In this second case, the role of media exposure to coronavirus-related content is highlighted as an intervening variable that has a direct negative effect on the accuracy of responses to visual and auditory WM tasks. Specifically, the value relating to the effects of exposure to media relating to COVID-19 on the percentage of correctness of the visual working memory responses is greater (β = 0.22, *p* < 0.005) than that relating to the effects on the accuracy of responses to auditory n-back tasks (β = 0.31, *p* < 0.03).

## 4. Discussion

In the current study, the relationships between media exposure, state anxiety, and visual and auditory WM during the COVID-19 pandemic, in a sample of healthy Italian adults, were examined. More specifically, the purpose of this study was twofold: (a) to evaluate the influence of high levels of media exposure and COVID-19 media exposure on levels of anxiety, and (b) to evaluate the effects of high levels of anxiety on the performance of both visual and auditory WM tasks. The findings of the present study show that media exposure to information related to COVID-19 is highly related to anxiety and subjective loneliness. High levels of social distancing, such as people who never or almost never meet others, or who do not go out of their home, do not show any correlations with anxiety and media exposure. High levels of subjective loneliness, instead, show correlations with both anxiety and media exposure, meaning that it is not living in objective social distancing situations that generates anxiety, but rather how those situations are processed (perceived loneliness). The direct relationship between media exposure and anxiety and loneliness seems to indicate that continuing to stay informed does not lead to anxiety reduction, but on the contrary it may amplify it [43]. According to the Social Compensation/Hypothesis (SCH) [8], increases in interactions through the media should compensate for reductions in in-person interactions. Moreover, as pointed out by the theory of media dependency [10], it is precisely during a serious social disturbance that a high need to find information, as well as to maintain and strengthen interpersonal relationships in the name of comparison and mutual support, emerges [11]. From the results of the present study, the SCH theory has to be handled with caution for several reasons: (a) during a time of crisis, people tend to increase their dependency on the media in an insistent and continuous search for increasingly accurate and updated information, in order to make adequate decisions on protective behaviors; (b) the increase in dependency on the media does not decrease anxiety levels, but may increase them; (c) interactions through the media and exposure to media can lead to a decrease in working memory accuracy. As the path diagrams show, it is not the time of exposure to the media that generates anxiety; rather, it may be the opposite: very anxious people may expose themselves more to the media. As confirmed by Boursier et al. [15], isolation probably reinforced the individuals’ sense of loneliness, thus strengthening the need to be part of virtual communities, and prolonged access to social media during the pandemic risked further increases in anxiety, thereby generating a vicious cycle. Moderating variables that can affect the adaptation of people to crises must found in more complex models that take into account biological, behavioral, cognitive and emotional factors. In this context, the coronavirus pandemic represents an unusual opportunity to investigate whether communication via the media can effectively compensate for face-to-face interactions [16]. Although the psychological impact of lockdown and quarantine on people was previously documented [17], the present study shows that it can also have an impact on WM processes. This study also has some limitations. The study was cross-sectional and was only targeted at the early stage of the COVID-19 pandemic in Italy. A longitudinal study could be conducted to observe the impact of media at different stages of the pandemic. Other limitations could be the relatively small sample (101 participants) and working memory tasks being the only cognitive component. Moreover, this study was conducted in different provinces of Italy and the socio-cultural components may involve biases as compared to other locations and cultures. Aspects of Italian national culture that affect businesses and individual psychology include family, education, and attitudes and values [44]. Another limitation is related to the nature of social media samples in the present study; as specified above in the participant section, the participants were recruited online. As Lehdonvirta, Oksanen, Räsänen, and Blank [45] underline, in online surveys, the probability of a given population member ending up in the sample is not specified and is unknown; for this reason, online surveys provide non-probability samples. The use of non-probability samples in social and policy research needs to be treated with caution. As the authors of [45] suggest, surveys of this type can be used to show that certain characteristics or phenomena exist (have non-zero probability), and thus, can also be used as exploratory studies. In the present study, we used n-back performance and online survey data in combination and this may make it possible to reduce the burden of measurement errors. Future studies should provide much larger samples, be conducted cross-culturally, and examine additional subcomponents of executive functioning.

## 5. Conclusions

Two main conclusions can be drawn from the present paper: (1) it is not social distancing due to the COVID-19 pandemic that influences anxiety levels, but rather the way in which people perceive, comprehend, and interpret the social distancing, that creates subjective loneliness, and consequently increases anxiety levels; (2) social distancing and the consequent increase in media exposure can negatively influence not only anxiety but also working memory processes.

## Figures and Tables

**Figure 1 ijerph-18-09279-f001:**
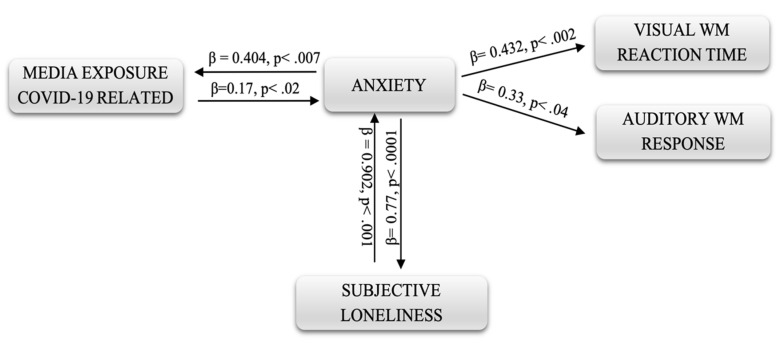
Path diagram.

**Figure 2 ijerph-18-09279-f002:**
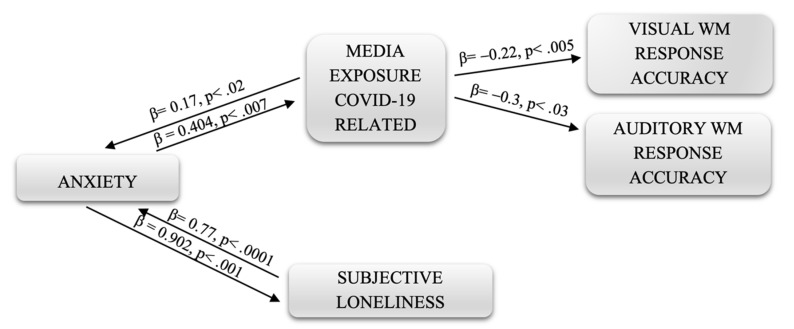
Path diagram.

**Table 1 ijerph-18-09279-t001:** Descriptive statistics of media exposure, COVID-19 media exposure, subjective loneliness, social distancing and anxiety status.

	Means	Standard Deviations
Media exposure	3.242	0.61
COVID-19-related media exposure	3.361	0.604
Subjective loneliness	2.112	0.796
Social distancing	2.709	0.617
Anxiety	50.198	13.58

**Table 2 ijerph-18-09279-t002:** Testing the mediation effect of subjective loneliness and social distancing between media exposure and anxiety.

	Media Exposure	COVID-19-Related Media Exposure	Subjective Loneliness	Social Distancing	Anxiety
Media exposure	-				
COVID-19-related media exposure	0.349 **	-			
Subjective loneliness	0.430 **	0.195 **	-		
Social distancing	0.007	0.123 **	0.188 **	-	
Anxiety	0.352 **	0.383 **	0.668 **	0.022	-

** *p* < 0.01.

**Table 3 ijerph-18-09279-t003:** Descriptive statistics of anxiety and working memory (WM) measurements.

	Means	Standard Deviations
Anxiety	50.19	13.58
Visual WM correct responses (VCR)	55.03	18.51
Visual WM reaction time (VRT)	0.02	0.42
Auditory WM correct responses (ACR)	48.18	22.71
Auditory WM reaction time (ART)	1.14	0.41

**Table 4 ijerph-18-09279-t004:** Pearson’s correlation between anxiety and working memory measurements.

	Anxiety	Visual WM Correct Responses	Visual WM Reaction Time	Auditory WM CORRECT Responses	Auditory WM Reaction Time
Anxiety	-				
Visual WM correct responses	0.04	-			
Visual WM reaction time	0.596 **	0.267 **	-		
Auditory WM correct responses	0.027	0.340 **	0.057	-	
Auditory WM reaction time	0.244 *	0.057	0.088	0.267 **	-

* *p* < 0.05, ** *p* < 0.01.

## Data Availability

Data are available on request to each of the authors.

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
