# Peer review of "The Influence of Media Exposure on Anxiety and Working Memory during Lockdown Period in Italy"

_ijerph, 2021, doi:10.3390/ijerph18179279_

Round 1
Reviewer 1 Report
This piece of research aims to examine the influence of media exposure on anxiety and working memory during lockdown period in Italy. This work is very interesting because it rmploys cognitive measures, which not only have less noise than traditional surveys, but also offer an integrative value to the study objective.
The greatest strength is the use of cognitive variables, as discussed above. However, I would like to suggest some studies for the theoretical framework on the influence of the mean in this context. A group of researchers have addressed a similar issue working in questions such as "I consider that I have correctly informed myself about Covid-19" in Italy and Spain. In these studies, feeling well informed about Covid-19 was related to fear of the economic impact of the crisis.
Murphy, M., & Moret-Tatay, C. (2021). Personality and attitudes confronting death awareness during the COVID-19 outbreak in Italy and Spain. Frontiers in Psychiatry, 12, 74.
A second study of the same group also related the same variable to groups of vulnerability during the covid-19, such as migrants:
Murphy, M., Lami, A., & Moret-Tatay, C. (2021). An Italian adaptation of the Brief Resilient Coping Scale (BRCS) and attitudes during the covid-19 outbreak. Frontiers in Psychology, 12.
Lastly, another interesting manuscript in the italian population (the facilitated and prolonged access to social media during the COVID-19 pandemic risked to further increase anxiety) is:
Boursier, V., Gioia, F., Musetti, A., & Schimmenti, A. (2020). Facing loneliness and anxiety during the COVID-19 isolation: the role of excessive social media use in a sample of Italian adults. Frontiers in psychiatry, 11.
On the other hand, the conclusions need to be rewritten in a simpler way. There is much more interesting information to be extracted than what the authors offer. They should emphasise the take-home message in simpler language for the average reader.
Minor points
-I do not understand the aim of figure 1 and 2
-Line 235, please correct Rsquare
Author Response
REFEREE 1
This piece of research aims to examine the influence of media exposure on anxiety and working memory during lockdown period in Italy. This work is very interesting because it employs cognitive measures, which not only have less noise than traditional surveys, but also offer an integrative value to the study objective.
The greatest strength is the use of cognitive variables, as discussed above. However, I would like to suggest some studies for the theoretical framework on the influence of the mean in this context. A group of researchers have addressed a similar issue working in questions such as "I consider that I have correctly informed myself about Covid-19" in Italy and Spain. In these studies, feeling well informed about Covid-19 was related to fear of the economic impact of the crisis.
Murphy, M., & Moret-Tatay, C. (2021). Personality and attitudes confronting death awareness during the COVID-19 outbreak in Italy and Spain. Frontiers in Psychiatry, 12, 74.
A second study of the same group also related the same variable to groups of vulnerability during the covid-19, such as migrants:
Murphy, M., Lami, A., & Moret-Tatay, C. (2021). An Italian adaptation of the Brief Resilient Coping Scale (BRCS) and attitudes during the covid-19 outbreak. Frontiers in Psychology, 12.
Lastly, another interesting manuscript in the italian population (the facilitated and prolonged access to social media during the COVID-19 pandemic risked to further increase anxiety) is:
Boursier, V., Gioia, F., Musetti, A., & Schimmenti, A. (2020). Facing loneliness and anxiety during the COVID-19 isolation: the role of excessive social media use in a sample of Italian adults. Frontiers in psychiatry, 11.
Reply
Thank you for your suggestions. We added Boursier, V., Gioia, F., Musetti, A., & Schimmenti, A. (2020) and Murphy, M., & Moret-Tatay, C. (2021) studies to enrich our theoretical framework. We added them in the enclosed paper both in the introduction and discussion sessions.
On the other hand, the conclusions need to be rewritten in a simpler way. There is much more interesting information to be extracted than what the authors offer. They should emphasize the take-home message in simpler language for the average reader.
Reply
Thank you. We re-write the conclusions
Minor points
-I do not understand the aim of figure 1 and 2
Reply
Thank you. We deleted them.
-Line 235, please correct Rsquare
Reply
Thank you. We corrected them.

Reviewer 2 Report
Abstract: Please include more specifics as to population sample and the analysis or methods that were used to analyze memory functions.
Introduction: Good use of psychological context regarding individual behavioral differences before and after the pandemic.
Good use of very recent literature on the influence of social and digital interactions on individuals during the pandemic.
Good explanation of background and purpose/focus of study.
Materials and Methods:
There is very thorough description of the participant pool including demographics, geographical locations, working status, and occupation which helps to provide context for the study.
Line 133 - Please give names or identify the two questionnaires that were adapted in addition to the sources provided. If these refer to the STAI and WM tests, please clarify this. While the other questionnaires seem to be explained below, there needs to be some more specifics regarding their names or from where they were adapted in the introduction paragraph.
Section 2.5 – which IRB board (from which institution)?
Results: Good use of tables and diagrams to provide visuals of the results.
Discussion and Conclusion: Good discussion of results with fair assessment of limitations. It should also be considered the cultural components of Italy compared to other locations and cultures when looking at limitations of the study. Although this is touched upon when looking at self-isolating individuals, it should also be considered from a cultural perspective.
Author Response
REFEREE 2
Abstract: Please include more specifics as to population sample and the analysis or methods that were used to analyze memory functions.
Reply
Thank you. We included more specifics as the population sample and the method to analyze memory functions.
Introduction: Good use of psychological context regarding individual behavioral differences before and after the pandemic.
Good use of very recent literature on the influence of social and digital interactions on individuals during the pandemic.
Good explanation of background and purpose/focus of study.
Reply
Thank you
Materials and Methods:
There is very thorough description of the participant pool including demographics, geographical locations, working status, and occupation which helps to provide context for the study.
Reply
Thank you
Line 133 - Please give names or identify the two questionnaires that were adapted in addition to the sources provided. If these refer to the STAI and WM tests, please clarify this. While the other questionnaires seem to be explained below, there needs to be some more specifics regarding their names or from where they were adapted in the introduction paragraph.
Reply
Thank you. We specified the names of the two questionnaires
Section 2.5 – which IRB board (from which institution)?
Reply
Thank you. We specified the name of the institution of the IRB.
Results: Good use of tables and diagrams to provide visuals of the results.
Reply
Thank you
Discussion and Conclusion: Good discussion of results with fair assessment of limitations. It should also be considered the cultural components of Italy compared to other locations and cultures when looking at limitations of the study. Although this is touched upon when looking at self-isolating individuals, it should also be considered from a cultural perspective.
Reply
Thank you, we added this limitation in the discussion section

Round 2
Reviewer 1 Report
Thank you for following my sugestions
Author Response
Dear Academic Editor,
my replies are:
still have a few additional comments and concerns that need addressing. These are detailed below.
1. Given that the study used data collected from social media platforms, it would be beneficial to add discussion about the nature of social media samples, which are typically non-probability based and self-selected. See, e.g.: Lehdonvirta, V., Oksanen, A., Räsänen, P., & Blank, G. (2021). Social media, web, and panel surveys: using non-probability samples in social and policy research. Policy & internet, 13(1), 134-155
Reply
Thank you for this observation. In the Discussion section we added this issue and discuss it (red highlighted).
Regarding study procedure: Please add information about the timeline of WM data collection. Once the participants were contacted via Skype, Teams, or Zoom, and instructed about the second phase, how long was the www.cognitivefun.net website available for the participants to complete the visual n-back and auditory n- back tests? What was the average completion time of the tests?
Reply
Thank you. In the method section (2.3.4) we added all the details of the n-back tests (red highlighted).
English language and style are fine, pending minor spell checks and formatting. For instance, please unify spelling of "COVID-19", using capitalized letters, as per journal style.
Reply
Thank you. We unified spelling of COVID-19 in all the text and check it for minor spell and formatting.
